# Design and Construction of High-Performance Long-Span Steel Transfer Twin Trusses Applied in One Hospital Building in Hong Kong

**Xiao-Kang Zou [1], Yi Zhang [2], Yao-Peng Liu [3,*], Liang-Cheng Shi [2] and Daniel Kan [1]**

1    Structures Research Hub, Building Construction Department, China State Construction Engineering (Hong Kong) Ltd., Hong Kong, China
2    China State Construction Engineering (Hong Kong) Ltd., Hong Kong, China
3    NIDA Technology Company Limited, Hong Kong Science Park, Hong Kong, China
*    Correspondence: yaopeng.liu@connect.polyu.hk

**Abstract:** Long-span steel trusses are increasingly used in high-rise buildings to replace reinforced concrete thick transfer plates due to their light weight but high load-bearing capacity. To support multiple stories above the steel transfer trusses, a comprehensive method based on second-order direct analysis has been applied for design optimization of long-span steel transfer trusses in one hospital redevelopment project in Hong Kong. In the project, several 35 m long-span steel transfer trusses are adopted at the 3rd to 5th floors to support the upper 15-story reinforced concrete structure. Innovative technologies such as integrated global and local optimization and integrated design and construction have been explored and made to achieve better uniformity and compatibility in structure. In particular, twin trusses with better structural performance, less fabrication cost and ease of constructability are studied and finally adopted in primary trusses to replace the original scheme of single trusses. The optimal scheme has brought both cost and time saving in fabrication, construction, operation and maintenance stages.

**Keywords:** steel truss; twin truss; single truss; optimization; second-order direct analysis

## 1. Introduction

Long-span steel trusses are increasingly used in high-rise buildings to replace reinforced concrete thick transfer plate due to light weight but high load-bearing capacity. To support multiple stories above the steel transfer truss, the top chord is subjected to high compression, leading to an out-of-plane stability problem. The effective length of the top chord for buckling check is changed from the construction stage to the final stage. Due to insufficient restraints provided in the construction stage, the actual effective length is unknown. Thus, the second-order direct analysis method is more preferred for practical design of long-span structures.

Designing long-span steel structures requires special considerations due to their span-to-depth ratio, complexity in supporting conditions, and various load patterns. These structures are typically used in large open spaces such as stadiums, airports, exhibition halls, and industrial buildings. Generally speaking, engineers need to consider these key factors for long-span steel structures.

(1)    Structural form: It significantly affects the structural performance. The choice of form depends on the factors such as the span length, the height of the structure, and the type of loading it is subjected to. Common forms include trusses, arches, cable-stayed structures, and space frames.

(2)  Material selection: Steel is commonly used due to its high strength-to-weight ratio and durability, but other materials such as concrete, timber, and composites can also be used. The material needs to be carefully selected based on the specific requirements of the project.

(3)  Design loads: Long-span steel structures are typically subjected to a variety of loadings, including dead loads, live loads, wind loads, and seismic loads. The loads need to be carefully considered and analyzed to ensure that the structure can withstand them safely.

(4)  Connections: The connections between different structural elements are critical to the performance of long-span steel structures. The connections need to be designed to transfer loads efficiently and safely, and to accommodate any movements or deformations that may occur.

(5)  Fire protection: Long-span steel structures are vulnerable to fire, and adequate fire protection measures need to be put in place to ensure the safety of occupants and the stability of the structure. This may include fire-resistant coatings, fireproofing, and sprinkler systems.

(6)  Maintenance: Long-span steel structures require regular maintenance to ensure their continued performance and safety. Maintenance may include inspections, repairs, and replacement of components as necessary. In summary, designing long-spann steel structures requires careful consideration of a range of factors.

In Hong Kong and Mainland China, the second-order direct analysis (SODA) method is quite commonly adopted in design offices for conventional structural design under static loads [1,2]. SODA considers P-$\Delta$ and P-$\delta$ effects as well as initial imperfections; therefore, it is widely adopted for design of long-span structures not only in the final completed stage, but also in construction stages.

Second-order nonlinear analysis is an advanced method for stability design of steel structures that offers several advantages over traditional first-order linear analysis. This method takes into account the nonlinear effects of structures under loading, which can significantly affect their stability and safety. Second-order analysis is particularly useful for tall or slender structures, where the effects of second-order deformations can be significant. One of the main advantages of second-order analysis is its ability to accurately predict the behavior of structures under different loading scenarios. This method can account for the interaction between different structural components, including beams, columns, and braces, and the effects of both axial and bending moments. This allows engineers to design structures that are more efficient, cost-effective, and safe. Another advantage of second-order analysis is its ability to identify potential failure modes and design alternatives. By analyzing the behavior of structures under different load cases, engineers can identify critical load paths, weak points, and areas that require reinforcement. This information can be used to develop alternative design solutions that optimize the structural performance and reduce the risk of failure. In a word, second-order analysis is an essential tool for modern structural engineering, providing engineers with a powerful tool to design safe and efficient steel structures.

High-strength steel is increasingly used in high-rise buildings and long-span structures. Extensive research on high-strength concrete-filled steel tubular columns [3–5] as well as steel joints [6,7] has been conducted in the past decade. More concerns on stability problems in steel design may be raised with the use of high-strength steel. Thus, SODA is promoted in many design codes as an advanced method for structural analysis and design.

For high-rise buildings, the steel transfer truss should be designed with adequate stiffness to replace the conventional concrete transfer plate. In this paper, a comprehensive method based on second-order direct analysis has been applied for optimization design of long-span steel transfer trusses in one hospital redevelopment in Hong Kong [8,9]. In the project, several long-span steel trusses are located at the 3rd to 5th floors, with an area of 35 m × 49 m, a total height of 9.6 m and also a total original steel weight of 2037 ton, as shown in Figure 1. As the site location is near an existing hospital building and an

important museum, an optimal design by using twin trusses is proposed to pursue the best construction quality and reduce the implication to the environment. This new scheme has better structural performance, less fabrication cost and ease of constructability. A detailed study for the twin trusses has been conducted and finally adopted in the main trusses to replace the original single-truss scheme.

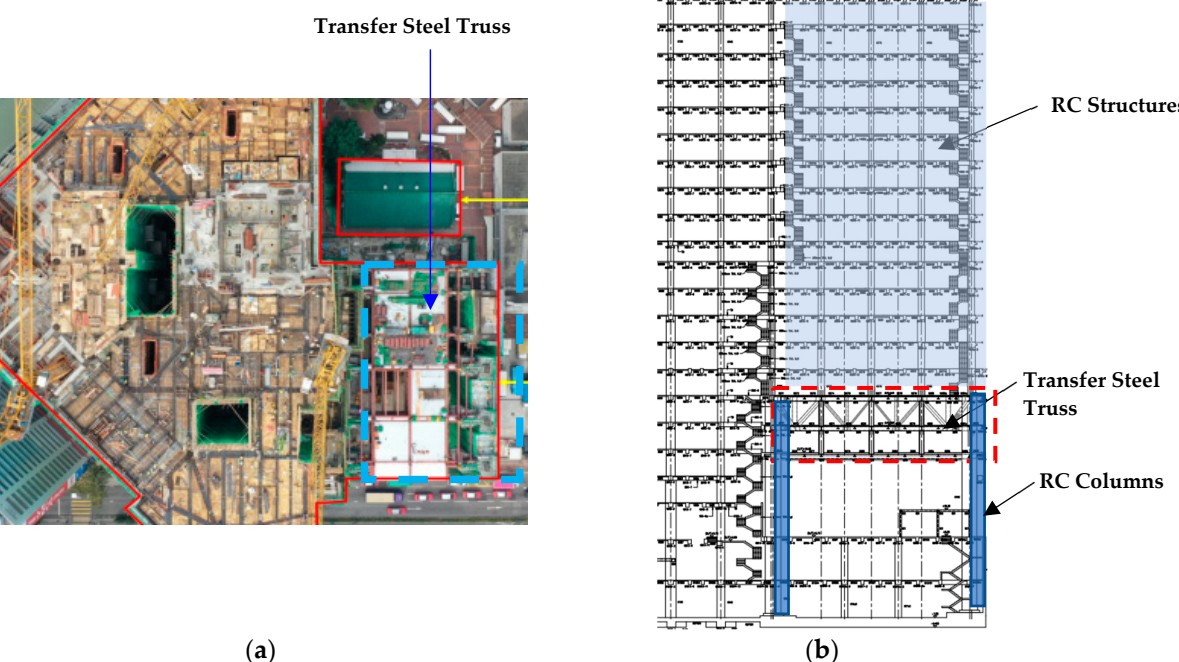

**(a)** **(b)**

**Figure 1.** Location of steel trusses in one Hong Kong hospital. (**a**) Aerial view of the project in construction; (**b**) Elevation of superstructure.

Novel systematic optimization techniques had been developed for elastic and inelastic structures under static and dynamic loadings and extensive optimization studies had been carried out [10–15]. Some of the optimization techniques had been applied to real building projects to improve design quality successfully and had resulted in certain economic benefits. In this paper, the optimal work mainly focuses on the steel transfer trusses of tall buildings. The optimal design maintains the original structural safety but brings great potential benefits to the project such as cost and time-saving, technical advantages, construction productive advantages and public nuisance reduction respectively.

## 2. Second-Order Direct Analysis

SODA has been well researched, and its concept has been specified in many modern design codes such as EC3 [16], AISC-LRFD [17], CoPHK [9] and GB50017 [18]. The initial imperfections in both global frames and local members are the key factors which should be included in the analysis process of SODA. The global and local imperfections can be considered by adjusting coordinates of each node and setting initial bowing of each member, respectively, based on eigen-buckling modes prior to SODA. The former can be easily handled in the analysis procedures while the latter should be supported by robust and reliable beam–column elements allowing for initial bowing such as the PEP element [19] and the curved stability function [20]. Bai et al. [21] proposed a tapered element with consideration of the initial imperfection. Tang et al. [22] proposed a co-rotational framework to enhance numerical efficiency with consideration of the distributed member loads. In this project, the flexibility-based beam–column element with initial geometric imperfection (hereafter called FBMI) proposed by Du et al. [23] for the second-order inelastic analysis is adopted for structural optimization. The FBMI element is briefly

introduced as follows for completeness. The local member basic forces and deformations of the FBMI element can be seen in Figure 2.

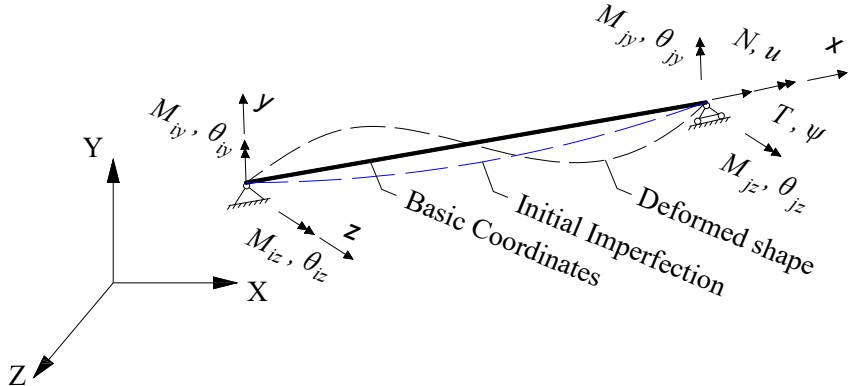

**Figure 2.** Force vs. displacement relations in basis system.

The FBMI element with initial geometrical imperfections is derived with the Hellinger–Reissner (HR) variational principle. The HR variational principle in terms of the displacement field $u$ and the stress field $\sigma$ is expressed as

$$\Pi_{HR}(\sigma, u) = \int_{\Omega} [\varepsilon(x, y, z)\sigma - \chi(\sigma)]d\Omega + \Pi_{ext}(u) \tag{1}$$

The stationary of the HR potential can be imposed by taking the first variation of Equation (1) regarding the displacement field $u$ and the stress field $\sigma$, and then letting it to zero as

$$\delta\Pi_{HR}(S, u) = \delta_S\Pi_{HR} + \delta_u\Pi_{HR} = 0 \tag{2}$$

The equilibrium and compatibility equations are obtained using Equation (2) and given in Equations (3) and (4), respectively.

$$\delta_u\Pi_{HR} = 0 \tag{3}$$

$$\delta_S\Pi_{HR} = 0 \tag{4}$$

The expanded form of the equilibrium equations in Equation (3) can be further written as

$$\delta_u\Pi_{HR} = \int_L S^T \left\{ \begin{array}{c} \delta u' + v'\delta v' + w'\delta w' + v_0'\delta v' + w_0'\delta w' \\ \delta v'' \\ -\delta w'' \\ \delta\psi' \end{array} \right\} dx - \overline{P}^T \delta D = 0 \tag{5}$$

where $u(x)$, $v(x)$ and $w(x)$ are the translational displacement components; $\psi(x)$ is the torsional angle about the $x$ direction; $v_0(x)$ and $w_0(x)$ are the initial bow imperfections; $S(x)$ is the sectional stress resultants; $\overline{P}$ is the end forces at the specified boundaries; $D$ is the end displacements $\{u\theta_{iz}\theta_{jz}\theta_{iy}\theta_{jy}\psi\}^T$. A half-sine function is widely recommended by design codes for the initial imperfections $v_0(x)$ and $w_0(x)$ as

$$v_0(x) = \delta_{0,y}\sin\frac{x}{L}; \ w_0(x) = \delta_{0,z}\sin\frac{x}{L} \tag{6}$$

in which $\delta_{0,y}$ and $\delta_{0,z}$ are the amplitudes of the initial bow imperfections at mid-span along local y- and z-axes, respectively, and can be found in Table 1.

More details of the flexibility matrix of the FBMI element can be referred to Du et al. [23]. The element accounting for initial imperfection has been incorporated into the software NIDA [24], which is used here for optimization design such that the effective length method is no longer required.

**Table 1.** Values of member initial bow imperfection used in design.

| Buckling Curves (CoPHK [9]) | $\frac{e_0}{L}$ Used in the Second-Order Direct Analysis |
|---|---|
| $a_0$ | 1/550 |
| a | 1/500 |
| b | 1/400 |
| c | 1/300 |
| d | 1/200 |

In this paper, the second-order elastic analysis with the first plastic hinge approach is adopted for design purposes. It means that the linear material model is used in the analysis. Not more than one plastic hinge is allowed within the design load combinations, and material non-linearity is not considered. It is normal design practice for the analysis to be elastic assuming no plastic moment redistribution. Once the first plastic hinge is detected, the analysis process is terminated.

## 3. Steel Truss Design Optimization

### 3.1. Steel Truss Schemes

Twin-truss schemes are adopted in three primary trusses A-A, B-B and C-C to replace the original single trusses. Note that the headroom does not decrease, as shown in Figure 3. To minimize member sizes, additional bracings are added at the ends of the three primary trusses to reduce the high stress without change of the original architectural functions.

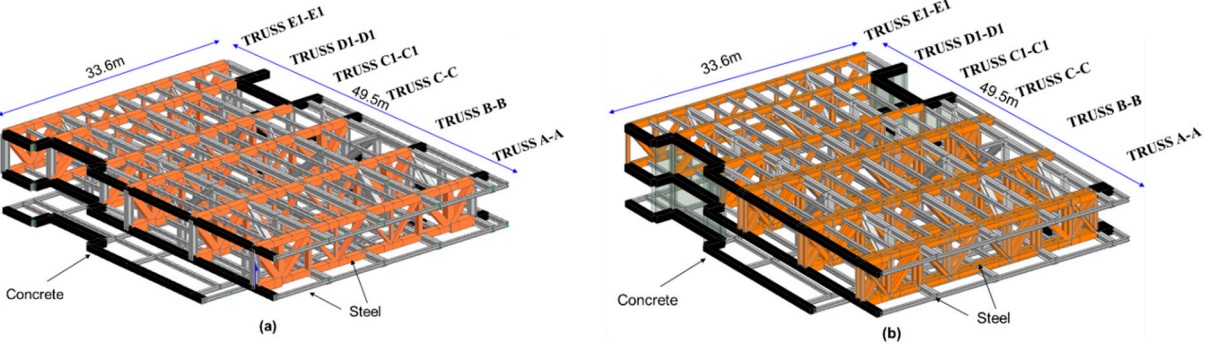

**Figure 3.** Overall view of the steel trusses. (**a**) Original design scheme; (**b**) Optimal design scheme.

In this project, the software NIDA complied with design codes such as CoPHK (2011) and GB50017 (2017) for second-order direct analysis is adopted. The details of analysis theory can be referred to the literatures [25–28].

### 3.2. Steel Section and Section Type

Open H-Sections are also used to replace the original box sections. From Figure 4 and Table 2, the number of box sections used is reduced, while the number of H-sections used is increased. It indicates that the quality and time of construction are more effectively controlled because the H-sections are more easily fabricated and installed.

**Table 2.** Comparison of member section types.

| Section Type | Original Design | | Optimal Design | |
|---|---|---|---|---|
| | Types | % | Types | % |
| Box | 17 | 81% | 6 | 25% |
| H-Section | 4 | 19% | 18 | 75% |
| Total | 21 | | 24 | |

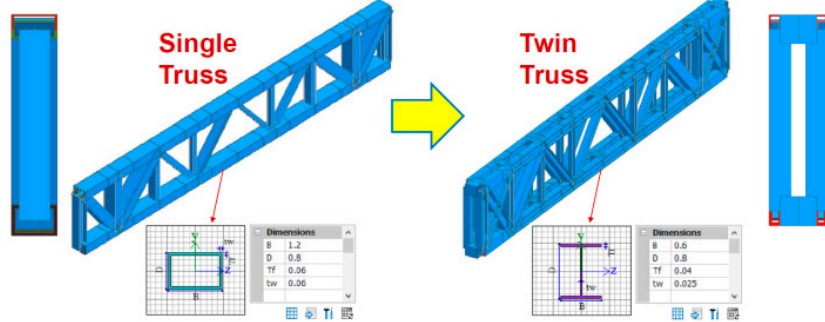

**Figure 4.** Original single truss vs. optimal twin truss.

### 3.3. Steel Grade

It is further found from the following Table 3 that steel tonnage with a grade of S460 is reduced from 1459 ton for the original design to 955 ton for the optimal design, while steel tonnage with a grade of S355 is increased from 578 ton to 801 ton. It indicates that the optimal design has brought better welding quality control. The distribution of S460 in the original design and the optimal design is shown in Figure 5 for easy reference.

**Table 3.** Comparison of steel grade used.

| Steel Grade | Original Design (Ton) | | Optimal Design (Ton) | |
|---|---|---|---|---|
| S460 | 1459 | 72% | 955 | 61% |
| S355 | 578 | 28% | 801 | 39% |
| Total | 2037 | 100% | 1756 | 100% |

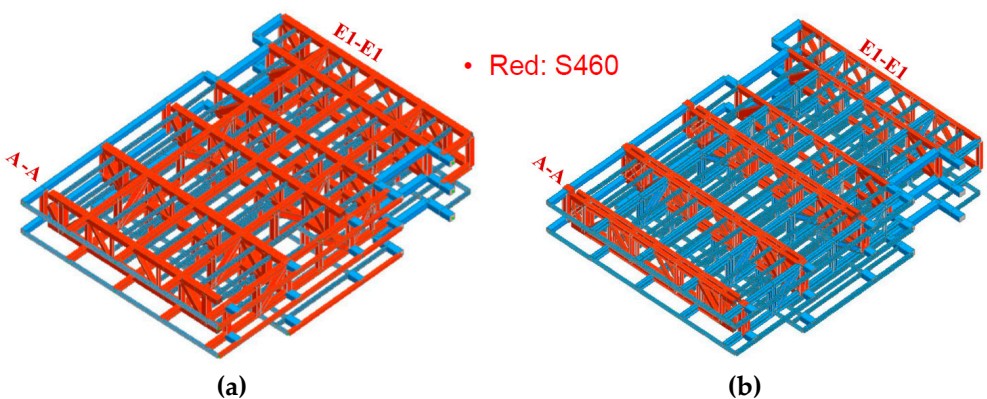

**Figure 5.** Distribution of high-grade steel S460. (**a**) Original design; (**b**) Optimal design.

### 3.4. Joint Connection Details

Joint connection details between the longitudinal truss with the transverse truss are also changed from the original complicated bolted and welded connection to the simple welded connection, as shown in Figure 6. In the optimal design, the joint connection is simpler and the welding procedure is also more convenient, which presents benefits to effectively control construction quality.

It should be pointed out that the optimal connection is a typical pinned connection without moment transfer, while the original joint detail is in between the pinned and semi-rigid connection because of extensive use of stiffeners. The optimal connection has a clearer load path.

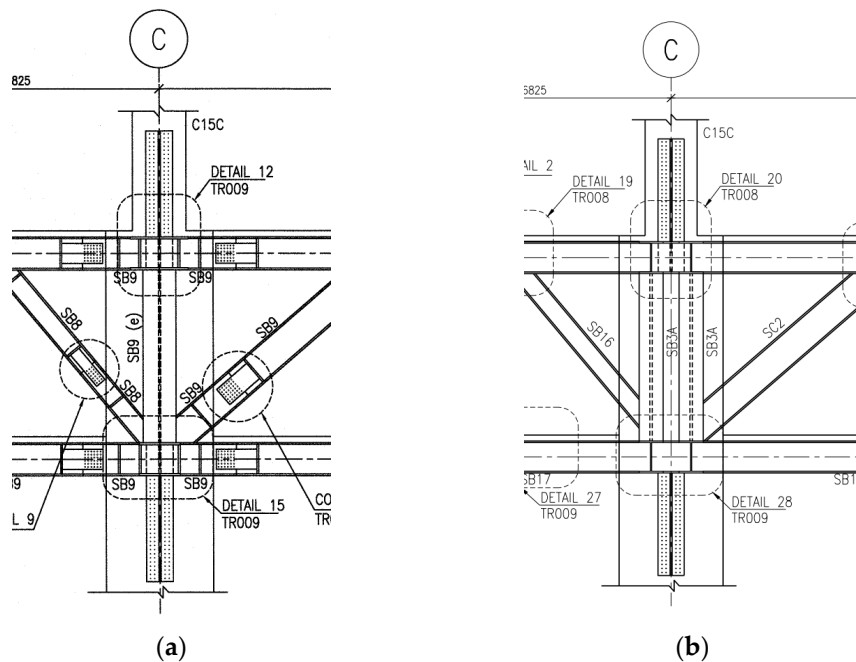

(**a**)                                             (**b**)

**Figure 6.** Original and optimized joint connection details. (**a**) Original design: welded and bolted; (**b**) Optimal design: welded.

## 4. Benefits from Optimal Design

### 4.1. More Resalable Weight Distribution

Original and optimal steel weight distributions are shown in Figure 7a,b, respectively. The primary truss weight is 1128 ton for the original design and 1017 ton for the optimal design. The comparison indicates that, for the optimal design, the ratio of primary truss steel tonnage over total tonnage (i.e., 55%) is close to that for the original design (i.e., 58%). Furthermore, it demonstrates the primary trusses also take an important role in the overall structure of the optimal design.

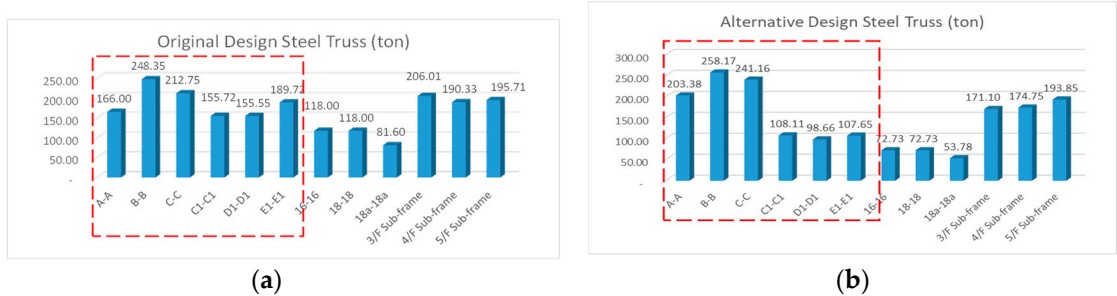

(**a**)                                             (**b**)

**Figure 7.** Distribution of steel weight. (**a**) Original steel truss scheme; (**b**) Optimal steel truss scheme.

### 4.2. Reduction of Steel Truss Self-Steel Weight and the Use of Z Plate

Steel with a normal grade is readily available in the market and may be obtained directly through a steel stockholder. However, two to three months may be required for material ordering for Z-grade steel. Hence, if a lot of Z plates are used, their order could not be flexible during construction stage. Therefore, high-grade and Z plate are not preferred for use in large quantities for a practical project.

Due to the use of twin trusses replacing the original single truss, the thickness of the chord member can be reduced from 80 mm to either 40 mm or 60 mm, mainly resulting in a decrease in the usage of Z-grade steel and the volume of welding. It is found from the following Table 4 that the steel truss 281 ton self-weight is preserved from the original 2037 ton to the final 1756 ton, achieving a decrease of 14%. The steel weight of high-grade

S460 is also significantly reduced from the original 1459 ton to the final 955 ton; the tonnage of Z plate has been reduced from the original 1421 ton (70%) to the current 748 ton (43%). It indicates that the implication caused by steel tonnage of high-grade and Z plate to construction could be reduced and easily controlled once future design is changed.

**Table 4.** Comparison of steel weight corresponding to steel grade and Z plate.

| Self-Weight (Ton) | Original Scheme | | Optimal Scheme | |
|---|---|---|---|---|
| | **Tonnage** | **Ratio = W/Total** | **Tonnage** | **Ratio = W/Total** |
| S460 weight—$W_{S460}$ | 1459 | 72% | 955 | 61% |
| S355 weight—$W_{S355}$ | 578 | 28% | 801 | 39% |
| Z-pate weight—$W_{Z\text{-plate}}$ | 1421 | 70% | 748 | 43% |
| Total weight | 2037 | 100% | 1756 | 100% |

*4.3. Uniformed Deflection for All Primary Trusses*

Table 5 and Figure 8 show primary truss deflections for the original and optimal design schemes, respectively. In summary, three findings can be observed as follows.

**Table 5.** Deflections for all primary trusses.

| Truss | Span | Limit (Span/360) | Deflection (Original Design) | Deflection (Optimal Design) |
|---|---|---|---|---|
| A-A | 32.9 m | 91.3 mm | 43.3 mm | 40.6 mm |
| B-B | 32.9 m | 91.3 mm | 50.7 mm | 47.5 mm |
| C-C | 32.9 m | 91.3 mm | 50.5 mm | 46.6 mm |
| C1-C1 | 28.3 m | 78.6 mm | 31.5 mm | 37.5 mm |
| D1-D1 | 28.3 m | 78.6 mm | 24.1 mm | 36.7 mm |
| E1-E1 | 33.6 m | 93.3 mm | 23.9 mm | 42.0 mm |

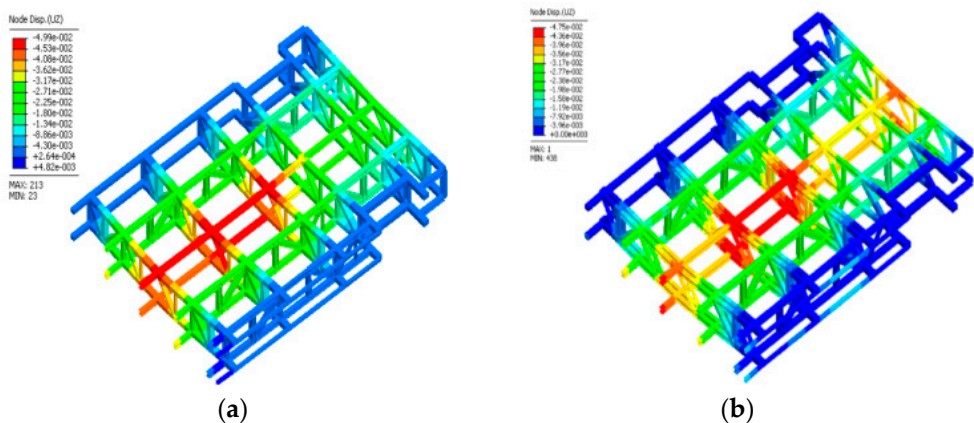

(a)　　　　　　　　　　　　　　　　(b)

**Figure 8.** Comparison of vertical displacements from NIDA. (**a**) Original design; (**b**) Optimal design.

(1)　For the optimal design, the deflections of Trusses A-A, B-B and C-C, supporting heavy loads from the superstructure, are essentially the same as those of the original design. Their discrepancies are only within 0.9 mm to 2.7 mm (i.e., 1.7–6.2%).

(2)　For the optimal design, the deflections between the trusses are close to each other, i.e., 36.7 mm to 47.5 mm with a difference of 10.8 mm. For the original design, the truss deflections vary from 23.9 mm to 50.7 mm with a difference of 26.8 mm. It means that, in the alternative design, the integrity of trusses and the concrete slab are better than that in the original design. That is, the optimal design has better uniformity and compatibility.

(3)　The truss deflections are only about half of the code allowable limits, which can fulfil the code deflection requirements.

In general, the maximum vertical displacement between the original and the optimal design are very close. Thus, the optimal design does not affect the overall structural behavior of the whole building, leading to minimal adverse effects on other structural elements such as RC columns and RC beams.

### 4.4. Increase of Overall Capacity

Based on the second-order analysis results, the maximum vertical deflection of the original design truss is 49.9 mm under full loading condition. The alternative design scheme enhance the overall stiffness of the truss and the maximum vertical deflection is reduced to 47.5 mm when under the same loading condition. This can be interpreted the overall capacity of the steel truss has been enhanced.

### 4.5. Reduction in Column Reactions

From Table 6, it can be observed that the column reactions obtained from the optimal design are less than those from the original design. It indicates that the optimal design has better force distribution and no implication to parent structure, compared with the original design.

**Table 6.** Comparison of column total reactions.

|  | Original Design | Optimal Design | Difference |
| --- | --- | --- | --- |
|  | $F_z'$ (kN) | $F_z$ (kN) | $(F_z - F_z')/F_z$ (%) |
| Vertical Load (Dead+SDL+Finishes) | 354,632 | 351,751 | −0.81 |
| Live Load | 133,398 | 127,855 | −4.16 |

### 4.6. More Reasonable Overall Utilization

It is found from the following Figure 9 that the member utilizations are less than the factor of 1.0, and their distributions are more reasonable. It indicates that all steel members are safe, and the material is full of use in the optimal design scheme. In Figure 9, the vertical axis shows the percentage of the total members, while the horizontal axis shows the utilization factor from 0.1 to 1.0. If the utilization factor is greater than 1.0, it means the members at the region are structurally inadequate.

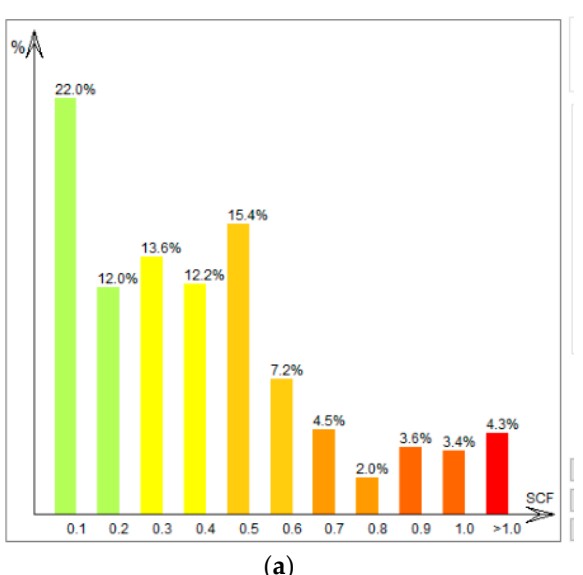
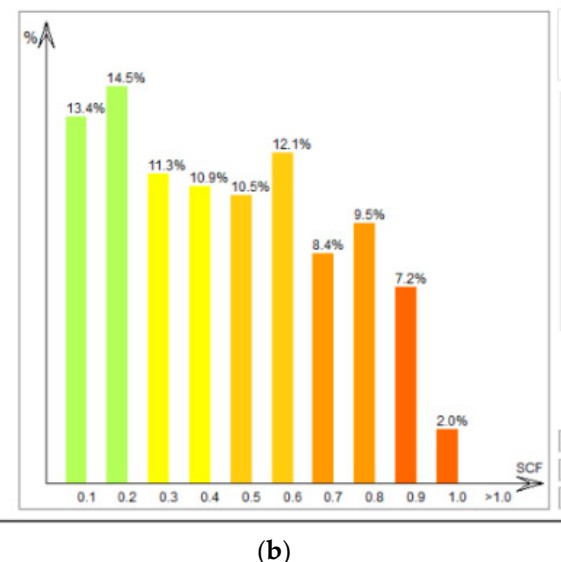

**Figure 9.** Utilizations of Steel Members. (**a**) Before optimization; (**b**) After optimization.

*4.7. Time Saving*

The prime consideration is to secure the whole of the works to be completed by the completion date. To achieve this objective, the following factors should be taken into account. First, one key factor affecting the construction period of the steel truss is on-site assembly, in particular welding. It is commonly known that on-site welding is a relatively time consuming and risky activity. Therefore, reduction in the amount of on-site welding can shorten the working time. Having regard to this key factor, the optimization work also focused on reduction in the thickness of the plate for the build-up sections. Detailed welding reduction and saving can be found in the following Table 7 of Section 4.7.

**Table 7.** Comparison of welding volume.

|  | Thickness of Plate (mm) | Weld Volume per mm (mm$^3$/mm) | Difference (%) |
|---|---|---|---|
| Original Design | 80 | 3200 | - |
| Optimal Design | 60 | 1800 | −43.75 |

Secondly, another key factor is that 80 mm thick steel plates rarely stock in the market and are a long-lead item by as much as three months. Thus, the steel plate should be replaced by other steel plates which are more available in the market. The expected time for delivery can then be assured. By virtue of the design optimization, the time required for the weld joint can be minimized; hence, the whole works to be completed on time can be secured.

*4.8. Other Advantages*

4.8.1. Reduction in Environmental Pollution

The site is near an existing hospital and a museum. Thus, the welding should be reduced by the optimal design scheme adopted. Further, aerial infection should reduce as well as implications to the surrounding hospital and museum, since the gas emission during welding would be decreased.

4.8.2. Improvement of Construction Quality Control

Most truss chord members have been changed from box sections to H-sections. Regarding the maintenance inspection of structural steel work in the future, it can be easy for the Inspector/Engineer to inspect the interior of chord members with H-sections. Hence, a warning of undesirable deterioration can be found early. In addition, the quality of the welding connection would be increased for the smaller volume.

4.8.3. Construction Productivity Advantages

In considering welding connections, thinner steel plates require a smaller volume of welding at connections for a full penetration butt weld, as shown in Table 7. By comparing the most common steel plate thicknesses (i.e., 80 mm for the original design and 60 mm for optimal design), the volume of the weld can be reduced by 43%.

Moreover, the testing failure on the welding joint for 60 mm plates is much lower than that for the weld joint of 80 mm. The changes in plate thickness can reduce the time for rectification works for such weld joint defects.

4.8.4. Increase in Headroom

As shown in Figure 10, the headroom of 3/F at the both end of truss A-A, B-B and C-C is reduced from 6.2 m to 5.8 m. It means that the new design scheme provides 400 mm more headroom than the original design.

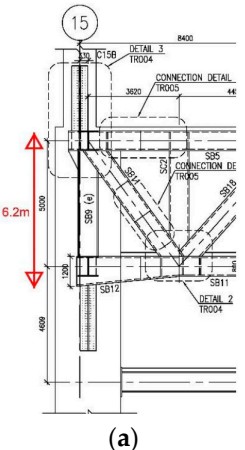
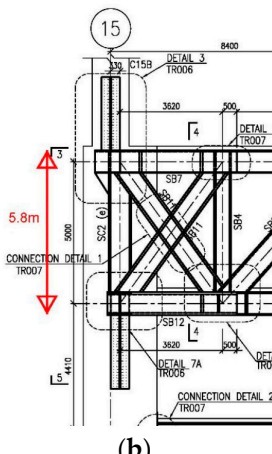

(**a**)  (**b**)

**Figure 10.** Increase of headroom at 3/F. (**a**) Original single truss scheme; (**b**) Optimal twin truss scheme.

### 4.8.5. Reduce Risk of Welder's Occupation Health

By changing most of the box section to H-section in the new design, the number of weld joints in overhead position is minimized. This reduces the risk of welder's occupation health.

### 4.8.6. Reduce Risk of Fire

The on-site welding is reduced in the new design. As a results, the fire risk related to welding work is also reduced.

### 4.8.7. Reduce in Smell Nuisance

Owing the reduction of on-site welding works, the smell nuisance to the public which generated during the welding process is also reduced.

### 4.8.8. Reduce in Traffic Impact

The number of traffic for the delivery of pre-fabricated steel truss elements to the Site is reduced due to the total weight of steel trusses is reduced in the new design. As a results, the traffic impact to the nearby public is also reduced.

## 5. Conclusions

In this paper, an optimal design based on the second-order direct analysis (SODA) method for long-span steel trusses is presented, which is feasible, safe, more cost-effective and demonstrates better structural performance. SODA considers P-Δ and P-δ effects as well as initial imperfections; therefore, the conventional effective length method for stability design is no longer needed. This method brings safer and more economical design with many other advantages.

Innovative technologies (such as the integrated global and local optimization, the integrated design and construction) have been explored and made to achieve better uniformity and harmony in structure. In particular, twin trusses with better structural performance, less fabrication cost and ease of constructability are studied and finally adopted in primary trusses to replace the original single trusses. The optimal scheme has brought both cost and time saving in fabrication, construction, operation and maintenance stages. Specifically, the overall weight of steel trusses has been reduced by 14% (i.e., 281 ton), the steel tonnage with high-grade S460 is reduced by 35% (i.e., 504 ton), and the volume of welding is reduced by 44% (i.e., 1400 mm$^3$/mm), leading to greater construction efficiency and less environmental pollution and carbon emissions.

**Author Contributions:** Conceptualization, X.-K.Z.; methodology, Y.-P.L.; validation, D.K.; writing—original draft preparation, X.-K.Z. and Y.-P.L.; writing—review and editing, Y.Z.; visualization, D.K.; project administration, L.-C.S. All authors have read and agreed to the published version of the manuscript.

**Funding:** This research received no external funding.

**Data Availability Statement:** Not applicable.

**Conflicts of Interest:** The authors declare no conflict of interest.

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
