# Peer review of "Design and Construction of High-Performance Long-Span Steel Transfer Twin Trusses Applied in One Hospital Building in Hong Kong"

_buildings, doi:10.3390/buildings13030751_

Round 1
Reviewer 1 Report
the paper is well structured. The topic is extremely interesting and above all current.It is advisable to refer to the English language in some sentences in the conclusions. However, the paper can be presented in this form.
Reviewer 2 Report
The research idea is good and suitable for the journal. The paper can be published after making the following modifications.
1- All symbols within the text must be in italics.
2- What does QTY in Table 2 mean? The meaning should be explained below the Table.
3- Figure 5 is not mentioned throughout the text.
4- In Figure (7-a), the first column, A-A, has the number 166, but in Figure was 66, please correct it.
5- The capital and small letters should be reviewed throughout the paper.
6- The research needs a major language review.
7- Through my review, the title should be changed to: “Design and Construction of High-Performance Long Span Steel Twin Trusses at the Redevelopment of Hong Kong Kwong Wah Hospital”
Reviewer 3 Report
This paper studied a comprehensive method based on second-order direct analysis method has been applied for optimization design of long-span steel transfer truss in the Redevelopment of Hong Kong Kwong Wah Hospital (KWH) – Phase 1. The numerical and theoretical investigations were conducted. However, there are still some questions, which need to be answered before the publication.
1, what is the material model used in numerical analysis? The linear model or the nonlinear model? More details on this issue should be added.
2, the differences in vertical displacements between the original and optimal designs are not very clear. More discussions are required.
3, Fig. 9. What is the longitudinal coordinates?
4, Section 4.6. is it possible to give the data and figure to clarify the time saving?
5, Section 3.4. The applicable of the simple weld connection should be proved, rather than weld and bolt.
Round 2
Reviewer 2 Report
It can be accepted for publication now.
Reviewer 3 Report
Thank you. This research paper could be accepted now.